# Programming of Vascular Dysfunction in the Intrauterine Milieu of Diabetic Pregnancies

**DOI:** 10.3390/ijms19113665

**Published:** 2018-11-20

**Authors:** Nada A. Sallam, Victoria A. C. Palmgren, Radha D. Singh, Cini M. John, Jennifer A. Thompson

**Affiliations:** 1Department of Physiology and Pharmacology, Libin Cardiovascular Institute of Alberta, Calgary, AB T2N 4N1, Canada; nada.sallam@ucalgary.ca (N.A.S.); victoriapalmgren@gmail.com (V.A.C.P.); radha.singh1@ucalgary.ca (R.D.S.); cini.john@ucalgary.ca (C.M.J.); 2Children’s Hospital Research Institute; University of Calgary, Calgary, AB T2N 4N1, Canada; 3Department of Pharmacology and Toxicology, Faculty of Pharmacy, Cairo University, Cairo 11562, Egypt

**Keywords:** gestational diabetes, endothelial dysfunction, developmental programming

## Abstract

With the rising global tide of obesity, gestational diabetes mellitus (GDM) burgeoned into one of the most common antenatal disorders worldwide. Macrosomic babies born to diabetic mothers are more likely to develop risk factors for cardiovascular disease (CVD) before they reach adulthood. Rodent studies in offspring born to hyperglycemic pregnancies show vascular dysfunction characterized by impaired nitric oxide (NO)-mediated vasodilation and increased production of contractile prostanoids by cyclooxygenase 2 (COX-2). Vascular dysfunction is a key pathogenic event in the progression of diabetes-related vascular disease, primarily attributable to glucotoxicity. Therefore, glucose-induced vascular injury may stem directly from the hyperglycemic intrauterine environment of GDM pregnancy, as evinced by studies showing endothelial activation and inflammation at birth or in childhood in offspring born to GDM mothers. This review discusses potential mechanisms by which intrauterine hyperglycemia programs dysfunction in the developing vasculature.

## 1. Prevalence and Pathophysiology

Gestational diabetes mellitus (GDM) is defined as glucose intolerance resulting in hyperglycemia that is first discovered during pregnancy. Universal screening for GDM by an Oral Glucose Tolerance Test (OGTT) administered between 24 and 28 weeks of gestation is part of routine antenatal care in North America [1]. New diagnostic criteria [2] were established in 2010 by the International Association of Diabetes and Pregnancy Study Groups (IADPSG), taking into account results from the Hyperglycemia and Adverse Pregnancy Outcome (HAPO) Study showing adverse perinatal outcomes to occur with maternal hyperglycemia below levels previously diagnostic of GDM. According to these new criteria, worldwide prevalence is 18%, ranging from 9 to 25% depending on the population studied [3]. High pre-pregnancy body mass index (BMI) is a major risk factor for GDM [4], and for this reason the obesity epidemic has spurred a parallel increase in the frequency of GDM. In North America, the rise in obesity was highest among women of reproductive age [5], with current rates at 30% in vulnerable populations [6]. Another important contributor to the increased prevalence of GDM over past decades is the trend of advanced maternal age, another significant predictor of GDM [7]. These trends have made GDM one of the most common complications of pregnancy in North America, which is now projected to become a widespread public health problem as the epidemiological transition continues in the developing world.

GDM typically develops in the second trimester when maternal insulin secretion fails to compensate for the progressive loss of insulin sensitivity, which serves to meet the glucose requirements of the growing fetus. This failure to maintain glycemic control in the pregnancy-induced state of insulin resistance is thought to unmask an underlying maternal predisposition. In support of this, women with a history of GDM have up to 50% higher risk for developing type 2 diabetes [8] and are reported to exhibit pancreatic β-cell dysfunction [9], a contributor to glycemic dysregulation during GDM pregnancy [10,11]. Loss of glycemic control in GDM may also stem from adipose tissue dysfunction characterized by unfettered lipolysis, inflammation and abnormal adipokine secretion. A systematic review by Bao et al., reported high leptin and low adiponectin levels in the circulation of first and second trimester pregnant mothers to be predictive of GDM development [12]. Circulating levels of pro-inflammatory cytokines have also been shown to independently predict the onset of GDM [13], and increased cytokine expression in subcutaneous adipose tissue of GDM women has been reported [14]. This pre-existing adipose tissue dysfunction and low-grade inflammation are thought to underlie the amplification of pregnancy-induced maternal immune activation observed in GDM women. While maternal immune activation in early gestation is necessary for establishment and maintenance of a healthy pregnancy [15], heightened inflammation in GDM indirectly influences pregnancy outcomes by compromising placental development and function [16]. Thus, GDM arises when the physiological stress of pregnancy unmasks a maternal predisposition that stems from a complex interaction between genetic and environmental factors.

## 2. Intrauterine Metabolic Milieu

The maternal metabolic dysregulation characteristic of GDM confers short and long-term risk to the developing fetus through adverse effects on the intrauterine environment. While maternal insulin does not cross the placenta, maternal glucose freely passes to the fetus along its concentration gradient and stimulates secretion of insulin from the fetal pancreas. Thus, maternal hyperglycemia subjects the fetus to a state of hyperinsulinemia and hyperglycemia, thought to be the major culprit for adverse perinatal outcomes associated with GDM [17]. Indeed, macrosomia (birth weight > 4000 g) occurs in 9% to 25% of GDM pregnancies [18,19] and is attributable to high glucose availability in conjunction with the anabolic effects of insulin, a major regulator of fetal growth. At birth, the hyperinsulinemic GDM fetus experiences an abrupt disruption in the maternal-fetal transfer of excess maternal glucose and for this reason is 5-times more likely to present with neonatal hypoglycemia [20]. Macrosomia also links to increased risk for unplanned cesarean section, shoulder dystocia, premature rupture of membranes and premature birth. The pathogenesis of GDM-induced macrosomia and related complications is now thought to be multifactorial, involving maternal metabolic derangements beyond hyperglycemia. Several studies have reported a positive relationship between birth weight and lipid levels in maternal or umbilical cord blood [21,22], which increase with loss of insulin-regulated inhibition of lipolysis. Further, high circulating leptin, another correlate to insulin resistance observed in GDM mothers [23], has been shown to contribute to fetal overgrowth [24]. These metabolic derangements underlying the pathogenesis and consequences of GDM are common to maternal obesity, which is present in the majority of GDM cases and independently predicts macrosomia [25]. Macrosomia due to intrauterine exposure to the metabolic milieu of GDM is associated with risks beyond the perinatal period, which are the focus of this review.

## 3. Cardiovascular Disease (CVD) Risk Factors in the Offspring

Babies born to mothers who were diabetic during pregnancy are prone to later-life development of the metabolic syndrome [26,27,28], a cluster of risk factors for cardiovascular disease (CVD) including obesity, insulin resistance and hypertension. Several of these studies show that the effects of diabetic pregnancy on offspring cardio-metabolic health manifest before the onset of adulthood, thus implicating intrauterine hyperglycemia as a contributor to the current epidemic of CVD risk among youth in the USA, Canada and other Western countries. Indeed, children between the ages of 5 to 18 who were born to diabetic mothers are reported to have impaired insulin sensitivity, high systolic blood pressure, elevated fasting triglycerides and BMIs within the overweight range [27,28,29]. While some studies report an attenuation of this relationship after adjustment for maternal pre-pregnancy BMI, overall, existing evidence implicates maternal hyperglycemia as an independent predictor of childhood CVD risk [29]. Adverse effects of maternal diabetes on offspring cardiovascular health are likely exacerbated by the co-existence of maternal obesity. Importantly, the association between maternal hyperglycemia and CVD risk factors in the offspring occurs independent of genetic susceptibility, as eloquently demonstrated in studies of the Pima, a North American indigenous tribe with the world’s highest rate of type 2 diabetes. Pima children aged 7–11 born to mothers who had diabetes during pregnancy had higher glycated hemoglobin HbA1c and systolic blood pressure compared to children born to mothers who developed T2DM after pregnancy [27]. Likewise, children and adolescents born to diabetic mothers had higher BMI and rates of diabetes compared to siblings born before the onset of maternal diabetes [30]. High BMI, circulating triglycerides and high-density lipoprotein (HDL) were reported in children aged 5–10 born to both treated and untreated GDM pregnancies [28], highlighting the need to consider GDM management in light of the mild degree of maternal glucose intolerance at which adverse offspring outcomes can occur.

## 4. Vascular Dysfunction: Human studies

Vascular dysfunction is a key pathogenic mechanism in the progression of CVD and common to the cardio-metabolic conditions that together constitute the metabolic syndrome. Very few studies have assessed indices of vascular function in human offspring born to GDM pregnancy. Table 1 summarizes findings in human studies. Tam et al. reported a positive association between umbilical insulin levels and the augmentation index, a measure of arterial stiffness, in teenagers born to GDM pregnancies [31]. Arterial stiffening is both a cause and consequence of hypertension and occurs when the mechanical properties of the arterial wall are altered by extracellular matrix changes brought on by inflammation or cyclic stress. Abnormal activation of extracellular matrix deposition after intrauterine exposure to hyperglycemia was evident by an increase in intima-media thickness in the abdominal aorta of 2–5-day-old infants born to GDM mothers, after adjustment for body weight [32]. Neo-intima formation is an initiating event in the atherogenic process, which frequently coincides with arterial stiffening and impaired relaxation. 

Impaired vasodilation due to reduced bioavailability of the major endothelium-derived relaxing factor, nitric oxide (NO), is a hallmark of vascular disease and significant predictor of adverse cardiovascular events [33]. There are several non-invasive measurements of endothelium-dependent dilation available for use in humans, but to date, have been rarely employed to investigate the relationship between GDM and vascular function in the offspring. Normal vasodilatory responses to acetylcholine, an endothelium-dependent vasodilator, were reported in 1–2-day-old infants of GDM mothers, as assessed by Laser Doppler imaging [34]. However, elevated circulating levels of cell adhesion molecules (markers of endothelial injury that initiate endothelial cell-leukocyte interactions and increase vascular permeability) have been observed by two studies in children born to mothers diagnosed with diabetes in pregnancy [26,35]. While the eventual development of vascular dysfunction secondary to programmed obesity and insulin resistance is expected, these data suggest that later-life vascular dysfunction stem directly from the intrauterine environment. In the adult setting, vascular injury due to transient exposure to hyperglycemia leads to lasting perturbations in vascular gene expression and function [36,37], thus it is reasonable to suppose that intrauterine hyperglycemic during critical periods of fetal vascular development programs persistent deleterious effects on vascular health.

## 5. Vascular Dysfunction: Animal Studies

Several studies report blunted relaxation in arteries from adult offspring born to diabetic pregnancy. In these studies, maternal diabetes was induced in pregnant rats by a single injection (35 to 50 mg/kg) of streptozotocin (STZ) at various times in gestation. Impaired dilation in response to acetylcholine (Ach), an endothelial nitric oxide synthase (eNOS) agonist, was consistently observed in the aorta and mesenteric arteries of offspring born to STZ-injected dams, albeit with a sex-specific effect suggesting protection in female offspring [17,53,54,55,56]. Results with respect to endothelium-independent dilation to sodium nitroprusside (SNP), a NO donor, were less consistent, with some studies showing a reduction and others failing to observe a difference. Along with increased systolic blood pressure, heightened pressor responses to phenylephrine were reported in adult offspring born to STZ-injected pregnant rats [57]. While one study observed normal ex vivo phenylephrine-induced contractile responses of isolated mesenteric arteries despite elevated blood pressure in male offspring from diabetic pregnancies [54], the majority of studies report heightened responses to vasoconstrictor agonists, phenylephrine and norepinephrine [17,53,55]. Further, enhanced contractile responses to electrical field stimulation [58] in the superior mesenteric artery of male offspring was reported by de Querioz et al. Overall, the above studies highlight vascular dysfunction characterized by blunted endothelium-dependent relaxation and heightened contraction in offspring from STZ-injected dams. 

Several findings implicate the endothelium as a key contributor to vascular dysfunction in offspring born to STZ-injected dams. For instance, differences in contractile responses in offspring from diabetic versus control pregnancies were abolished with endothelium denudation [55] and inhibition of eNOS failed to increase contractile responses to Angiotensin II in offspring from diabetic dams [59]. These results suggest a reduced ability of endothelial-derived agents to restrain vasoconstriction. Cyclooxygenase-2 (Cox-2), an enzyme expressed in endothelial cells that contributes to vascular tone via production of vasoactive prostanoids, is a potential mediator of programmed endothelial dysfunction. Increased expression of Cox-2 but not Cox-1 was observed in mesenteric arteries of male offspring [60]. Treatment of isolated vessels with indomethacin, a non-selective Cox inhibitor or NS-398, a specific Cox-2 inhibitor, normalized Ach-induced responses in offspring from diabetic dams but had no effect on offspring from normal pregnancies [53,60]. Inhibition of thromboxane A2 (TxA2) synthesis or TxA2 receptor antagonism similarly enhanced Ach-induced relaxation in offspring from diabetic pregnancies [53,60], suggesting that an imbalance in the production of dilatory vs contractile prostanoids underlies the contribution of Cox-2 to vascular dysfunction.

The above studies indicate that abnormal vascular reactivity due to underlying endothelial dysfunction presents in a sex-specific manner in offspring born to dams that were hyperglycemic during pregnancy. However, caution is warranted when interpreting these studies since all used STZ injection to induce hyperglycemia during early pregnancy. STZ is a cytotoxic drug that causes highly variable levels of hyperglycemia by destroying pancreatic β-cells. Since STZ has toxic effects on multiple organs and leads to drastic weight loss, it is possible that STZ injection in early pregnancy has adverse effects on developmental outcomes independent of hyperglycemia. The majority of studies cited above report that offspring born to STZ-treated dams are growth restricted at birth and fail to catch-up in the postnatal period. By cross-fostering a subset of pups, Holeman et al., demonstrated that early growth impairments in offspring born to STZ-injected dams were primarily attributable to the suckling period, suggesting that irreversible effects of STZ injection on maternal health negatively influence lactation performance [56]. Therefore, there is a need to develop new animal models of GDM to identify the mechanisms linking intrauterine hyperglycemia to CVD risk factors in the offspring. The pregnant mouse heterozygous for leptin receptor deficiency (Lepr^db/+^) is a potential tool to study GDM as it captures central features including higher adiposity, hyperleptinemia and glucose intolerance [49,61]. A recent study failed to observe glucose intolerance in pregnant Lepr^db/+^ females [62] suggesting that maternal adiposity or hyperleptinemia may play more prominent roles in developmental outcomes. Nonetheless, variations in experimental design such as maternal age at pregnancy and male genotype of mating pair may account for discrepancies in reported metabolic features of Lepr^db/+^ pregnancy. Moreover, stress responses may mask any subtle differences in glucose tolerance when measured by a glucose tolerance test in live mice. Lepr^db/+^ pregnancy appears to adversely affect metabolic health in the offspring, as shown by a report by Nadif et al. wherein wild-type male offspring born to Lepr^db/+^ dams exhibited lower insulin sensitivity [61]. One study found greater arterial stiffness, but no evidence of endothelial dysfunction in offspring born to Lepr^db/+^ dams [50]; however, more studies are needed to characterize the vascular phenotype of offspring born to Lepr^db/+^ pregnancies. It is important to note that endothelial dysfunction in offspring born to dams fed a high fat diet has been frequently reported, and is likely attributable to a number of maternal factors including insulin resistance [63]. In summary, it is highly probable that GDM has deleterious effects on the developing vasculature; however, current animal models make it difficult to discriminate the impact of maternal hyperglycemia from maternal obesity or other features of metabolically compromised pregnancy.

## 6. Mechanisms: Oxidative Stress

Reactive oxygen species (ROS) mediate pathways in the cell that control inflammatory responses, survival, differentiation and proliferation [64]. In fact, redox signaling plays a role in the establishment and maintenance of pregnancy. Gestational age-related increases in placental ROS production driven by increasing metabolic activity and oxygen tension act as a stimulus for placental angiogenesis and trophoblast differentiation and invasion [39]. Under this pregnancy-induced state of high oxidant load, redox homeostasis is maintained by up-regulation in the expression and activity of placental antioxidants. When the capacity of antioxidants to scavenge excess ROS are overwhelmed, oxidative stress ensues, exposing the immature cellular defense system of the fetus to high levels of placenta-derived ROS. A role for oxidative stress in developmental programming has been demonstrated by several studies showing that later-life health in offspring born to adverse pregnancy is rescued with antenatal interventions effective in mitigating maternal oxidant load [40,41].

It is well documented that oxidative stress is a key mechanism in the pathogenesis of endothelial dysfunction in adult-onset diabetes [33]. Hyperglycemia leads to the generation of excess ROS via multiple mechanisms including increased flux through the polyol and hexosamine pathways and protein kinase C (PKC) activation. Excess ROS impair endothelial function by depleting NO availability and cause oxidative damage to cellular DNA, lipids and proteins. Indices of oxidative damage are elevated in maternal, placental and fetal tissues from GDM pregnancies. For instance, malondialdehyde (MDA), a byproduct of lipid peroxidation, was elevated in maternal plasma, cord blood and placental tissue of GDM pregnancies [42,46,47]. Placental tissue extracted from GDM pregnancies had higher protein carbonyl content, a marker of protein oxidation, and exhibited greater release of 8-isoprostane, an indicator of lipid peroxidation [43]. While some studies show compensatory increases in antioxidant defenses in placentae and cord blood of GDM pregnancies, others report decreases in the expression and activation of endogenous antioxidants [43,44]. Overall, available data indicate that GDM creates a state of oxidative stress, exposing the developing fetal vasculature to the deleterious effects of excess ROS.

There exists evidence to suggest that the harmful effects hyperglycemia-induced oxidative stress on endothelial function in the offspring are present at birth. Chen et al. showed elevated mitochondrial superoxide production and increased markers of lipid and protein oxidation and DNA damage in human umbilical vein endothelial cells (HUVEC) isolated from GDM pregnancies [45]. This oxidative stress appears to be a causative factor in programming of abnormal phenotypic properties such as reduced proliferative and migratory capacities, impaired tubule formation and apoptosis in GDM-derived HUVEC and endothelial colony forming cells (ECFC) [65,66,67,68]. Sultan et al. showed that impaired proliferation in GDM-derived HUVECs was accompanied by increased intracellular ROS formation and restored with antioxidant treatment [69]. Another study showed high superoxide levels and reduced NO bioavailability despite increased endothelial NO synthase (eNOS) expression in GDM-derived HUVEC, suggesting eNOS uncoupling wherein dysfunction eNOS generates superoxide rather than NO [70]. However, Sáez et al. reported unaltered ROS generation under basal conditions in GDM-HUVEC despite increased eNOS expression and impaired wound healing [71]. High glucose treatment in cultured ECFCs isolated from cord blood of normal pregnancy reproduced the abnormal proliferative and migratory properties observed in GDM-derived cells [67]. Likewise, in vitro high glucose exposure increased markers of oxidative stress and enzymatic sources of ROS in HUVEC collected from normoglycemic pregnancy [72]. Thus, later-life vascular health in offspring born to GDM mothers may stem directly from glucose-induced oxidative stress in the intrauterine environment. 

## 7. Mechanisms: Inflammation

Oxidative stress is coupled with heightened inflammation, as there is a high degree of intersection between redox and inflammatory signaling pathways. Through activation of redox sensitive transcription factors, ROS regulate the gene expression of several pro-inflammatory molecules including cytokines, cell adhesion molecules (CAM), and Cox-2, a major mediator of vascular inflammation. Indeed, amplification of pregnancy-induced immune activation is common to GDM pregnancies. Increased circulating concentrations of pro-inflammatory cytokines including tumor necrosis factor-α (TNF-α), interleukin-6 (IL-6) and monocyte chemoattractant protein-1 (MCP-1) have been reported in women diagnosed with GDM [73,74]. A recent study by Lobo et al., found that in the blood of GDM patients there was a higher percentage of cytotoxic natural killer (NK) cells and greater production of TNF-α by Treg cells, compared to glucose-tolerant controls [75]. While dysfunctional maternal adipocytes are likely sources of circulating pro-inflammatory molecules in GDM pregnancies, the placenta is thought to play an important role in mediating maternal inflammation, as it is both a target and source of cytokines. During healthy pregnancy, each trimester is associated with a distinctive pattern of placental cytokine expression necessary for the various stages of placentation. This tightly regulated balance in placental expression of pro-inflammatory and anti-inflammatory mediators is disrupted in GDM, as demonstrated by studies showing aberrant cytokine production [51,76]. In ex vivo perfused placental cotyledons, the increase in TNF-α production coinciding with late-gestation was exaggerated in GDM pregnancies, although in both control and GDM cotyledons 94% of TNF-α was released to the maternal side and only 6% released to the fetal side [13]. Further, Aye et al. reported high maternal BMI to be associated with elevated levels of MCP-1 and TNF-α in maternal serum and p38-MAPK and signal transducer and activator of transcription 3 (STAT3) in the placenta, while levels of inflammatory markers in umbilical blood were unchanged [77]. These data suggest that the placenta serves as a barrier to limit fetal exposure to pro-inflammatory signaling molecules derived from the placenta or maternal tissues. Therefore, heightened activation of maternal and placental inflammatory pathways in GDM pregnancies likely influences the fetus through effects on utero-placental perfusion and placental nutrient delivery, given the role of inflammatory pathways in placental development. In fact, aberrant nutrient transport in the GDM placenta is well documented [48]. In cultured human trophoblasts, IL-6 and TNF-α treatment led to an up-regulation of amino acid transporters [78], while IL-1β inhibited the insulin signaling pathway and insulin-stimulated amino acid transport [52]. 

It is widely acknowledged that the vascular injury common to diabetic patients is attributable to glucose-induced inflammation in vascular cells. Therefore, inflammatory signaling in the fetal vasculature may be directly responsive to intrauterine hyperglycemia in GDM pregnancies. A pro-inflammatory phenotype was recently observed in HUVEC collected from GDM pregnancies, as evident by heightened monocyte adhesion and increased CAM levels after TNF-α treatment [38]. Likewise, healthy HUVEC exposed to high glucose increase expression of CAMs, intercellular adhesion molecule-1 (ICAM), vascular cell adhesion molecule 1 (VCAM-1) and E-selectin [79]. West et al. reported higher circulating levels of VCAM-1 and E-selectin, along with higher BMI and blood pressure in 6–13-year-old children born to diabetic mothers [35]. Thus, intrauterine exposure to hyperglycemia may program a pro-inflammatory vascular phenotype that predisposes to vascular dysfunction and atherosclerosis. In the setting of adult-onset diabetes, even transient exposure to hyperglycemia has an enduring effect on vascular inflammatory gene expression through epigenetic mechanisms.

## 8. Mechanisms: Epigenetics

GDM is a transient state of hyperglycemia and thus the recent recognition of the “metabolic memory” whereby transitory periods of hyperglycemia install persistent epigenetic-mediated changes in gene expression implores consideration of this phenomenon as an agent of fetal programming. Epigenetic processes involved in transcriptional regulation include methylation of cytosine residues at regions of DNA dense in CpG dinucleotide clusters and posttranslational modification of histone tails. The latter entails ubiquitination and acetylation of lysine, phosphorylation of serine and threonine and methylation of lysine and arginine, all of which regulate gene expression through conformational changes in chromatin, which in turn dictate accessibility of transcriptional activators or repressors. Of these posttranslational histone modifications, methylation of lysine and arginine residues is the most stable. Histone methylation has variable effects on gene activity dependent on the specific histone tail, the position of the residue within the histone tail and the degree of methylation. In contrast, hypermethylation and hypomethylation of CpG dinucleotides invariably results in gene repression and gene activation, respectively. Mounting evidence implicates DNA and histone methylation in the progression of diabetes-related vascular disease [37,80]. Thus, persistent dysregulation of gene expression owing to the effects of intrauterine hyperglycemia on methylation patterns may underlie later development of vascular dysfunction in postnatal life.

Study of methylation patterns in tissues collected from GDM pregnancies have provided some evidence to suggest a role for epigenetics in programming of vascular dysfunction. DNA methylation levels of ATP-binding cassette transporter A1 (ABCA1), a critical regulator of lipoprotein metabolism, were found to be altered in the placenta and cord blood of GDM pregnancies, the latter showing a negative correlation with fasting maternal triglycerides and glucose [81]. Lipid metabolism and cytokine-mediated signaling were also identified as top differentially DNA-methylated networks in placentae from a diet-induced model of GDM [82]. Two studies reveal aberrant methylation marks in vascular cells collected from cord blood of GDM pregnancies. Floris et al., demonstrated that the histone methyltransferase of histone H3 lysine 27 (H3K27), enhancer of zester homolog 2 (EZH2), was suppressed by high levels of the microRNA, miR-101, in GDM-HUVEC [66]. EZH2 is not only a target but also a transcriptional repressor of miR-101, which is known to disrupt endothelial function and vascular development (76). Inhibition of miR-101 restored indices of functional capacity such as tubule formation in GDM-HUVEC [66]. Overexpression of placenta-specific-8 (PLAC8) in GDM-ECFCs was attributable to hypomethylation at regulatory regions and correlated with the degree of maternal hyperglycemia [83]. Although the function of PLAC8 in ECFC is unknown, knockdown of PLAC8 corrected deficiencies in proliferative capacity of GDM-ECFC [83]. GDM-associated changes in DNA methylation patterns appear to be long lasting, as demonstrated by a study of 388 Pima native children that reported 48 differentially methylated CpG sites primarily in genes involved in the pathogenesis of obesity and diabetes [84]. Another cohort of children exposed prenatally to GDM showed differential methylation patterns in blood samples for a number of genes associated with cardiometabolic traits [85]. These included natriuretic peptide receptor-1 (NPRI), a gene associated with hypertension, and two genes involved in atherosclerosis. Further, two genes involved in the regulation of VCAM-1 expression were hypermethylated in concert with increased circulating levels of VCAM-1 [85]. The above studies highlight a role for transcriptional regulation via methylation in the development of phenotypic abnormalities observed in GDM-derived ECFC and HUVEC, as well as in the high CAM expression reported in young GDM offspring.

Small non-coding RNAs, miRNAs, have emerged as important mediators of gene expression patterns via posttranslational degradation or translational repression and interaction with other components of the epigenome. Various cellular functions are governed by miRNAs and several are implicated in normal cardiovascular development as well as the pathogenesis of diseases including diabetes, dyslipidemia and hypertension [86]. For instance, the endothelial cell-specific miRNA, miR-126, is a critical regulator of angiogenesis and vascular formation, as its inhibition resulted in defective vascular development evident by delayed angiogenic sprouting, leaky vessels, hemorrhaging and partial embryonic lethality [87]. Knockdown of miR-126 in endothelial microparticles interfered with reendothelialization after denudation of the endothelium, while impaired endothelial repair under hyperglycemic conditions was accompanied by a downregulation in miR-126 [88]. miR-155 modulates NO production through decreasing eNOS mRNA stability, and its inhibition resulted in increased NO availability and enhanced endothelium-dependent vasodilation [89]. In vascular smooth muscle cells, miR-145 is involved in differentiation and is downregulated during the phenotypic modulation that mediates neointimal formation [90]. While several miRNAs have been identified as markers of cardiovascular complications in the context of adult-onset T2DM [91], little is known regarding their role in programming of vascular dysfunction during hyperglycemic pregnancy. As discussed above, miR-101 was upregulated in GDM-HUVEC, while its inhibition restored endothelial cell function [66]. Since the histone methyltransferase, EZH2, is a target as well as transcriptional inhibitor of miR-101, increased miR-101 is associated with reduced EZH2 enrichment at the transcriptional regulatory site of miR-101 as well as reduced H3K27 trimethylation, resulting in a dysregulated feedback loop between EZH2 and miR-101 [66]. Further research is needed to identify the role of miRNAs in GDM-associated developmental outcomes and explore their potential as biomarkers for adverse programming.

## 9. Summary

Overall, there is a paucity of available data with respect to the relationship between GDM and vascular function in human offspring; however, there is some evidence to suggest that endothelial activation and aberrant extracellular matrix remodeling present shortly after birth in babies born to GDM pregnancies. Thus, later life risk for CVD in offspring born to GDM pregnancies may stem from direct effects of the intrauterine metabolic milieu on the developing vasculature. Hyperglycemia-induced oxidative stress and inflammation are likely culprits for vascular injury in the GDM fetus and may lead to persistent dysregulation of vascular gene expression patterns through epigenetic mechanisms (Figure 1). In support of this, a handful of studies demonstrate a role for oxidative stress and methylation-mediated transcriptional regulation in phenotypic programming of vascular cells collected from the umbilical cord. These data highlight glucose-mediated methylation changes as potential mechanisms by which endothelial dysfunction develops in animal offspring exposed to prenatal hyperglycemia. However, given the difficulty of modeling the spontaneous development of hyperglycemia that characterizes GDM, the overwhelming majority of animal studies have relied on the STZ model of type 1 diabetes. Future studies would greatly benefit from the development of new animal models that circumvent confounding influences of STZ-induced toxicity. Epigenetic programming during critical developmental windows may present unique therapeutic challenges; hence, future studies should aim to understand the functional relevance of differential epigenetic marks programmed by intrauterine hyperglycemia.

## Figures and Tables

**Figure 1 ijms-19-03665-f001:**
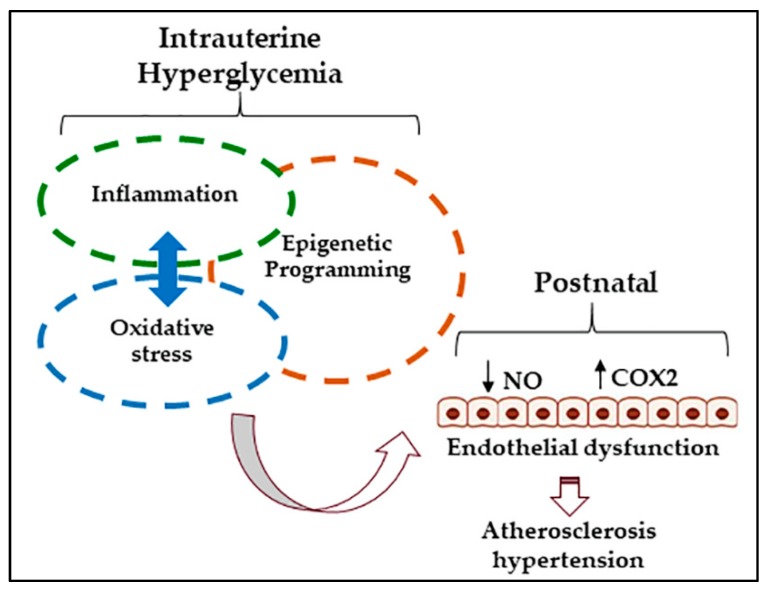
Mechanisms linking gestational diabetes mellitus (GDM) to vascular dysfunction in the offspring. Maternal hyperglycemia exposes the fetus to high levels of glucose. Deleterious effects of hyperglycemia are mediated largely through inflammation and oxidative stress. Highly interrelated pathways of inflammatory and redox signaling are persistently dysregulated after transient hyperglycemia due to transcriptional programming by DNA or histone methylation. Studies in endothelial colony forming cells (ECFC) and human umbilical vein endothelial cells (HUVEC) collected from GDM pregnancies suggest that intrauterine hyperglycemia directly programs vascular dysfunction by modulating methylation patterns and increasing oxidant load and inflammation. Chronic oxidative stress and inflammation lead to endothelial dysfunction characterized by decreased endothelium-derived dilators (NO) and increased contractile mediators (COX-2 prostanoids).

**Table 1 ijms-19-03665-t001:** Human evidence for programming of vascular dysfunction after GDM exposure.

Age Group	Examined Tissue/ParameTer	Phenotype	Reference
At birth	HUVEC	Reduced proliferation, tubule formation and migration capacity, increased apoptosis, higher superoxide production and markers of oxidative stress, decreased NO availability, increased monocyte adhesion, heightened inflammatory response	[38,39,40,41,42,43,44,45]
	ECFC	Reduced ECFC number and impaired proliferation, migration and tubule formation	[46,47,48]
	Cord blood	Increased markers of lipid peroxidation, change in DNA methylation status of genes involved in lipid metabolism and atherosclerosis	[49,50,51]
Infants (3–5 days old)	Abdominal aorta	Greater aortic intima-media thickness after adjustment for body weight	[30]
Children/adolescent (2–18 years old)	Blood	Increased triglycerides, increased cell adhesion molecules (CAM), change in DNA methylation pattern of genes involved in atherosclerosis and CAM expression	[25,26,35,52]
	Haemodynamics	High systolic blood pressure, positive association between umbilical insulin levels and augmentation index	[27,29,35]

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
