# Peer review of "Programming of Vascular Dysfunction in the Intrauterine Milieu of Diabetic Pregnancies"

_ijms, 2018, doi:10.3390/ijms19113665_

Round 1
Reviewer 1 Report
The manuscript is overall well written.
I just have some minor concerns:
- The Authors should expand the section on epigenetics, discussing the following topics (recently reviewed in J Cell Physiol. 2016 Aug;231(8):1638-44):
differentially expressed miRs were identified in the blood and in exosomes isolated from serum of healthy controls compared with patients with metabolic syndrome, type 2 diabetes, hypercholesterolemia or hypertension
a significant increase in circulating miR-126 has been detected in patients with acute myocardial infarction and angina whereas miR-126 downregulation has been reported in plasma from patients with diabetes, heart failure, or cancer.
- It is advisable to the Authors to incorporate a pictorial or cartoon representation of the proposed mechanisms in order to facilitate the comprehension and increase the overall impact of the manuscript.
Author Response
-The Authors should expand the section on epigenetics, discussing the following topics (recently reviewed in J Cell Physiol. 2016 Aug;231(8):1638-44):
Response: Thank you. We have added a paragraph focused on miRNA. In this section, we have highlighted that miRNAs are involved in the pathogenesis of several diseases and have described a few studies identifying a role for specific miRNAs in vascular disease/dysfunction. We also point to the study described in the preceding paragraph, wherein miR-101 was found to be involved in the dysregulation of HUVEC from GDM pregnancies.
-It is advisable to the Authors to incorporate a pictorial or cartoon representation of the proposed mechanisms in order to facilitate the comprehension and increase the overall impact of the manuscript
Response: Thank you. We very much agree and have added a figure to summarize the proposed mechanisms linking GDM to later-life vascular dysfunction in the offspring.
Reviewer 2 Report
To the authors
In this manuscript, the authors reviewed the association between maternal GDM and the offspring’s vascular function. This manuscript was well-written and summarized current findings. A few minor revisions are listed below.
1.The authors discussed the relationship between intrauterine environment and later-life vascular function. However, it is difficult for me to understand what effects occurred at birth and what did in adulthood. I am slightly confused. The authors should clearly organize what happened soon after birth and what happened after growth.
2.The authors mentioned high circulating leptin in GDM mothers. I think that it is useful to discuss the role of maternal leptin on the offspring’s vascular function mentioned in Pennington’ paper (PLoS One. 2016; 11: e0155377).
3.In line 50, “systemic” should read systematic.
4.In line 58-59, appropriate citations are required.
5.In line 88-89, appropriate citations are required.
6.In line 132-133, appropriate citations are required.
7.In line 137, is the subhead “Vascular reactivity” necessary?
Author Response
1. The authors discussed the relationship between intrauterine environment and later-life vascular function. However, it is difficult for me to understand what effects occurred at birth and what did in adulthood. I am slightly confused. The authors should clearly organize what happened soon after birth and what happened after growth.
Response: Thank you. We have modified our table to clearly differentiate the changes that have been observed at the following stages: birth, newborn, childhood/adolescence.
2. The authors mentioned high circulating leptin in GDM mothers. I think that it is useful to discuss the role of maternal leptin on the offspring’s vascular function mentioned in Pennington’ paper (PLoS One. 2016; 11: e0155377).
Response: We have included a discussion of this model and of the findings reported by the Pennington paper.
3. In line 50, “systemic” should read systematic.
Response: this was edited
4.In line 58-59, appropriate citations are required.
Response: citations were added
5.In line 88-89, appropriate citations are required.
Response: citations were added
6.In line 132-133, appropriate citations are required.
Response: citations were added
7.In line 137, is the subhead “Vascular reactivity” necessary?
Response: this subheading was removed